# Perceptual clustering in auditory streaming

**Nathanael Larigaldie** [1,2], **Tim Yates**[3], **Ulrik R. Beierholm** [1]*

**1** Durham University, Durham, United Kingdom, **2** Aarhus University, Aarhus, Denmark, **3** University of Birmingham, Birmingham, United Kingdom

* ulrik.beierholm@durham.ac.uk

## Abstract

Perception is dependent on the ability to separate stimuli from different objects and causes in order to perform inference and further processing. We have models of how the human brain can perform such causal inference for simple binary stimuli (e.g., auditory and visual), but the complexity of the models increases dramatically with more than two stimuli. To characterize human perception with more complex stimuli, we developed a Bayesian inference model that takes into account a potentially unlimited number of stimulus sources: it is general enough to factor in any discrete sequential cues from any modality. Because the model employs a non-parametric prior, increased signal complexity does not necessitate the addition of more parameters. The model not only predicts the number of possible sources, but also specifies the source with which each signal is associated. As a test case, we demonstrate that such a model can explain several phenomena in the auditory stream perception literature, that it provides an excellent fit to experimental data, and that it makes novel predictions that we experimentally confirm. These findings have implications not just for human auditory temporal perception, but for a wide range of perceptual phenomena across unisensory and multisensory stimuli.

## Author summary

Perceiving the world requires humans to organize perceptual stimuli according to the likely sources that generated them, requiring inference about these sources and their relationship with the generated stimuli. As an example, the sound of two different species of song birds should be segregated in order to better identify them. In this paper we utilize ideas from statistics to propose a way for the mind to do the allocation of stimuli to sources, that is not restricted by the number of stimuli, and that instead combines information dynamically as the complexity grows. We show that this not only qualitatively explains known phenomena in auditory perception, but can also quantitatively explain behavior, and leads to an experimental prediction that we confirm.

**Data availability statement:** The human behavioral raw data are available for both

experiments. Code, stimuli and data wrangling and analyses scripts for experiment 2 are also provided. Computational model predictions as well as the code for computational modelling and analyses scripts are also available. All can be found in the OSF repository: https://osf.io/ys94d/.

**Funding:** NL was funded through a Durham University Post-graduate grant. The funder had no role in study design, data collection and analysis, decision to publish, or preparation of the manuscript.

**Competing interests:** The authors have declared that no competing interests exist.

## Introduction

Ambiguity in perceptual systems is a blight for inference. When we hear two sounds sequentially, we may infer that they came from two different sources, e.g. birds A and B, or the same source repeated. A third sound is heard: Are the generating sources AAA, AAB, ABA, ABB, or ABC (see Fig 1 for the possible generative models)? By the time four, five and six sounds are heard the number of possible combinations reaches 15, 52, and 858. The ambiguity breeds to generate a combinatorial explosion, and yet the human auditory system is able to reliably allocate multiple sources of sound in complex, real-world situations. The characteristics of the signal are consistently associated with different sources, allowing us to keep track of a speaker's voice and the wail of an ambulance siren, separate from the noise of background traffic and falling rain.

Inferring which generative structure produced the observed stimuli is a general problem faced by the perceptual system. To perform such a task for simple stimuli the brain relies on causal inference [2,3], probabilistically estimating the most likely cause of the stimuli in the environment. This has been shown to be a good model of perceptual inference for ambiguous stimuli when they are in small numbers, e.g. two. However, increasing the number of stimuli causes the complexity of possible generative structures to rapidly increase (Fig 1), rendering a causal inference strategy that relies on enumerating all possible structures impossible.

However, an important realisation is that, given a specific set of a large number of stimuli, this process is essentially one of *clustering*, combining together stimuli that are similar while keeping separate from those dissimilar. This idea that perception involves clustering has a long history in the literature of Gestalt psychology, although not always expressed in those terms [4]. However, the brain would need to choose the right number of clusters and have a way to specify the prior expectations over clusters, which is difficult before even knowing the number of stimuli.

The key proposal in this paper is that the brain can perform this clustering process using a modern version of 'Occam's Razor', *non-parametric Bayesian clustering*. Specifically we will use the Chinese Restaurant Process [5]. With this approach, the number of clusters does not have to be pre-specified, as the algorithm adapts the number of clusters to the data. Likewise, there is no need for a large number of parameters to specify the prior expectations: a single meta-parameter ($\alpha$) specifies the degree of clustering.

Intuitively, the algorithm assigns customers (data points) at a fictional restaurant to different tables (groups and clusters) based on the number of customers already seated at the table. As the number of customers assigned to a table grows, it becomes more likely to be assigned further customers, making it less likely that new customers will be assigned a new table. This type of algorithm is renowned for allowing the complexity of the model to grow with the data set [5,6] as a larger number of clusters can emerge as the number of data points increases.

Non-parametric Bayesian inference has previously been used in cognitive and perceptual studies [7,8], but not to study the segmentation of perceptual cues.

To exemplify how human perception of large number of sources can be modeled by non-parametric Bayesian inference, we will present modeling and experimental results on auditory stream segregation.

### Auditory stream segregation

For several decades, the human ability to segregate sequential sounds into streams corresponding to sources has been investigated using simple sequences of either pure tones or more complex sounds (reviewed in [9]). The time interval between tones, their pitch difference and the duration of a sequence are among the factors that play an important role

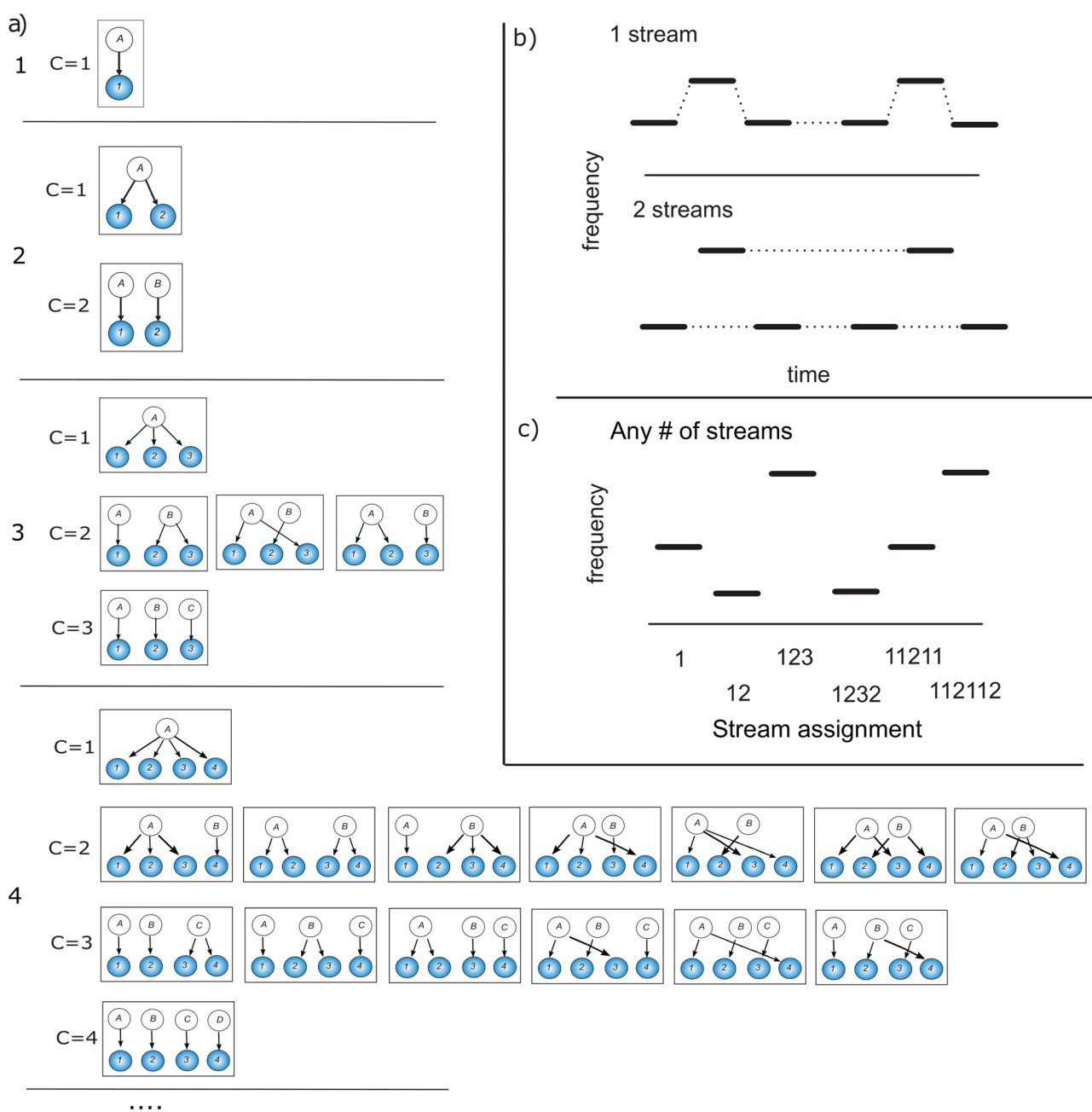

**Fig 1. a) Graphical illustration of the clustering problem in causal inference.** As the number of stimuli increases (1, 2, 3, 4, ...) the number ($C$) of potential causes increases at the same rate, while the number of combinations of causes that could have generated the stimuli increases according to the number of ways to partition a set of $n$ objects into $k$ nonempty subsets. It is easy to differentiate between the two potential generative structures when there are only two stimuli, but much harder when four stimuli can be created from fifteen different generative structures. b) Example of auditory tones being segregated into one or two streams, using 'galloping' stimuli similar to [1]. c) Example of a series of potential stimuli with a representative assignment of tones to the streams below. As each tone is presented, the observer reassigns the entire set of tones to streams (1−>12−>123 etc.). The brain has to decide how to assign each tone into an unknown number of streams, a type of clustering problem.

[1,10,11]: explanations of how the factors are used based on principles such as Gestalt laws and Occam's razor have been incorporated into the conceptual model of Bregman [12].

Descriptive models based on peripheral excitation [13], coherence of coupled oscillators [14] and cortical streaming modules [15] provide mechanisms to estimate the number of streams, but do not specify which sound is associated with which source. While some of the models are expandable to allow more sources to be inferred, it is not known if they would cope with the combinatorial explosion. Furthermore, Moore and Gockel [16] conclude from an extensive review of the literature that any sufficiently salient factor can induce stream segregation. This indicates that a more general model of inference is needed, that can incorporate any auditory perceptual cue and multiple sounds with different sources.

If ambiguity is a blight for inference, regularities in natural signals are the cure. Not all combinations of signal sources are equally likely – when perceptual systems generate a model of the world, we assume that they infer the most likely interpretation because the perceptual systems are optimized to the statistics of natural signals [17,18]. Bayesian inference has had considerable success in modeling many visual and multi-sensory percepts as a generative, probabilistic process [19–21]. Despite these successes, we still have no general, principled model of how the auditory system solves the source inference problem.

A Bayesian approach to auditory stream segregation (based on sequential sampling) has been used to model the dynamics of perceptual bistability [22] but assumes that only two percepts are possible. Turner [23] has developed methods of analyzing statistics of sounds based on Bayesian inference, and constructed a model to synthesize realistic auditory textures. While inference in the model can qualitatively replicate many known auditory grouping rules, the expected number of sources in the environment has to be specified. In a natural scenario an observer would have to infer the number of sources that are present in the environment due to the inherent uncertainty.

In our model the probability of many alternative stream configurations (given the input signal) are calculated and the percept generated corresponds to the most probable configuration. The probabilities are calculated using Bayes' rule to combine the likelihood of generating a signal given a postulated stream configuration, with the prior probability of sounds being associated with different sources. The likelihood and prior probability distributions are iteratively updated in a principled manner as information accumulates.

The forms of the distributions are presumably optimized to natural signal statistics: the likelihood distribution we use is based on consideration of the physical limitations of oscillators. However, the framework of the model allows formulations of multiple explanatory factors, such as those determined by Bregman [12] from psychophysics experiments, to be simply incorporated in the distributions. Furthermore, while the current study uses simple pure tones (replicating work by Bregman), the framework allows more complex cues from audition and other modalities to be used as long as their perceptual difference can be quantified.

## Model

Pure tones are the indivisible atoms of input to the model – each being assigned to just one sound source, or stream. Inspired by the work done on non-parametric priors [7,24,25] we assume the existence of an infinite number of potential sources, leading to a sequence of *tones* with pitch $f_1, f_2 \ldots$, onset time $t_1^{on}, t_2^{on} \ldots$ and an offset time, $t_1^{off}, t_2^{off} \ldots$ and the sound sources/streams that generated the tones are denoted by positive integers $S_1, S_2 \ldots$ We rename the sources when necessary so that the first tone heard will always be generated by source 1 (i.e. $S_1 = 1$), and a subsequent $i$th tone can be associated with source $[1 : max(S_1 \ldots S_{i-1}) + 1]$.

## Generative model

In order to introduce the likelihood function and priors we first explain the generative model. Given a source $S_i$ we assume that the frequency of tone $i$ is governed by physical constraints and statistical regularities of the source. If two sequential sounds with the frequencies $f_{i-1}$ and $f_i$ are produced by the same source, $S_i = S_{i-1}$, the pitch cannot change at an infinitely fast rate: to make an oscillator change its frequency discontinuously would require an infinite impulse of energy. A physical system will need time to change configuration so as to produce a different frequency and that the time to make that change depends on the difference in tone produced. If a tone of the same frequency is to be generated, no change in the system is required, while a larger tone difference requires more time. We assume that, all things being equal, a pure tone sound source is most likely to continue oscillating at the same frequency as it has in the past, and the probability decreases with $\Delta f = f_i - f_{i-1}$ but increases with $\Delta t = (t_i^{on} - t_{i-1}^{off})$ as the system needs time to change (on a scale of milliseconds). More specifically, we assume a normal probability distribution (see Fig 2):

$$p\left(f_i, t_i^{on} | S_i = S_{i-1}, f_{i-1}, t_{i-1}^{off}\right) = \frac{1}{\sqrt{2\pi\sigma^2}} exp^{-\frac{\left(\frac{\Delta f}{\Delta t}\right)^2}{2\sigma^2}} \tag{1}$$

where $\sigma$ is a constant, and $\Delta f$ is on a logarithmic frequency scale. We here assume that the observer has a perfect noise-free access to the generated auditory frequencies.

For successive sources, we assume that sources that have been active previously are more likely to be active again, but do not provide a limit to the number of sources that $N$ tones can be generated from. For simplicity, we also do not put in any temporal dependencies in the process (one could imagine that temporal gaps across seconds would make a source less likely to be active again). Concretely we assign the probability of a source $s$ generating the $i$th tone ($p(S_i = s)$) according to a Chinese Restaurant Process (CRP; [5]), which can be considered as an extension of Occam's rule.

If the number of tones previously assigned to source $s$ is given by $n_s = \sum_{k=1}^{i-1} \delta(s - S_k)$ then

$$p(S_i = s | S_1...S_{i-1}) = \frac{n_s}{(i1 + \alpha)} \qquad\qquad \text{if } n_s > 0 \tag{2}$$

$$p(S_i = s | S_1...S_{i-1}) = \frac{\alpha}{(i1 + \alpha)} \qquad\qquad \text{if } n_s = 0 \tag{3}$$

where $\delta$ is the discrete Kronecker delta function ($\delta(0) = 1$, but 0 elsewhere). Intuitively, the more tones have already been assigned to a source, the more likely that tone $i$ will also be assigned to it ("rich-get-richer" scheme). The parameter $\alpha$ specifies how quickly the accumulation of tones happens, and is a parameter we fit in our model (with the restriction $\alpha > 0$).

## Inference

The task of the observer is to infer the sources generating each of the tones, i.e. to find the sequence $S_1 S_2 S_3$ that maximize $p(S_1 S_2 S_3 | f_1 f_2 f_3, t_1^{on} t_2^{on} t_3^{on}, t_1^{off} t_2^{off} t_3^{off})$, as illustrated in Fig 1. For simplicity of writing we will refer to the properties of tone $i$ as $t_i$ in place of the set $(f_i, t_i^{on}, t_i^{off})$

As an example, we use a sequence of three tones $t_1, t_2, t_3$, for which the observer wishes to infer the likely sources $S_1, S_2, S_3$. Thus the probability $p(S_1, S_2, S_3 | t_1, t_2, t_3)$ that a sequence of three tones was generated by sources $S_1, S_2, S_3$, has to be calculated over the five possible combinations: $[S_1 = 1, S_2 = 1, S_3 = 1]$, $[S_1 = 1, S_2 = 1, S_3 = 2]$, $[S_1 = 1, S_2 = 2, S_3 = 1]$, $[S_1 = 1, S_2 = 2, S_3 = 2]$

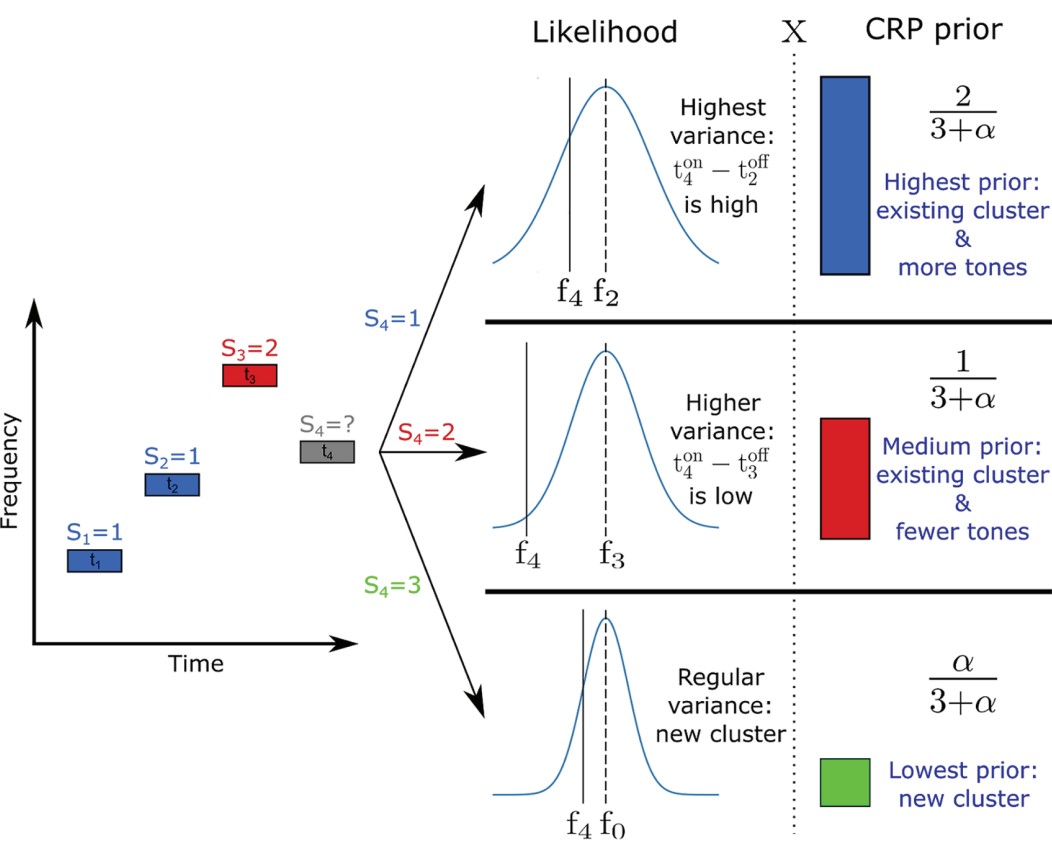

**Fig 2. Examples of the likelihood function and CRP prior for a 4th tone given that previous tones** $[t_1, t_2, t_3]$ **were generated by sources** $[S_1 = 1, S_2 = 1, S_3 = 2]$. This figure illustrates how $\Delta t = (t_i^{on} - t_{i-1}^{off})$ influences the probability of a tone being generated by different sources (as time distance increases, so does the capacity of the source to significantly change its oscillation frequency). It also shows how the CRP prior implements Occam's Razor by penalizing the probability of a new cluster, and has a "rich gets richer" property by favoring more populated clusters.

$2, S_3 = 2]$, $[S_1 = 1, S_2 = 2, S_3 = 3]$ corresponding to the five unique configurations of sources generating three sounds (see Fig 1a). Note that the first source is always assigned the value 1, the next different source is assigned 2, etc.

Bayes' rule relates each conditional probability (posterior distribution) to the likelihood $p(t_1, t_2, t_3 | S_1, S_2, S_3)$ of each configuration of sound sources generating the sequence of tones, by

$$p(S_1, S_2, S_3 | t_1, t_2, t_3) = \frac{p(t_1, t_2, t_3 | S_1, S_2, S_3) * p(S_1, S_2, S_3)}{p(t_1, t_2, t_3)} \tag{4}$$

where $p(S_1, S_2, S_3)$ is the prior probability of the particular configuration of sound sources, regardless of the frequency, etc. of the tones. In this form the equation states that the posterior depends on the joint likelihood of all the tones, multiplied by the joint prior over all sources. This is not always the most convenient way to formulate it, and we can instead write it in a manner that updates posteriors as each new tone arrives (see Fig A in S1 Text for a graphical representation of the model).

Assuming conditional independence of the tones and tone-source causality (i.e. $p(t_1|S_1, S_2, S_3) = p(t_1|S_1)$, and $p(t_2|S_1, S_2, S_3) = p(t_2|S_1, S_2)$ ) the posterior can be rewritten as

$$p(S_1, S_2, S_3|t_1, t_2, t_3) = \frac{p(t_3, t_2, t_1|S_1, S_2, S_3)p(S_1, S_2, S_3)}{p(t_1, t_2, t_3)}$$

$$= \frac{p(t_3|t_1, t_2, S_1, S_2, S_3)p(t_2|t_1, S_1, S_2, S_3)p(t_1|S_1, S_2, S_3) * p(S_3|S_1, S_2)p(S_2|S_1)p(S_1)}{p(t_1, t_2, t_3)}$$

$$= \frac{p(t_3|t_1, t_2, S_1, S_2, S_3)p(S_3|S_1, S_2) * p(t_2|t_1, S_1, S_2)p(S_2|S_1) * p(t_1|S_1)p(S_1)}{p(t_1, t_2, t_3)}$$

$$= p(t_3|t_1, t_2, S_1, S_2, S_3) * p(S_3|S_1, S_2) * p(S_1, S_2|t_1, t_2)/Z$$

(5)

where Z is a normalizing term. The last term $p(S_1, S_2|t_1, t_2) \propto p(t_2|t_1, S_1, S_2)p(S_2|S_1) * (t_1|S_1)p(S_1)$ is the posterior generated from the first two tones. The last two terms can be considered together as the prior for the third source, allowing us to use an iterative approach to the inference.

After each tone we grow the tree of possible source sequence (e.g. 11 → 111 and 112), by multiplying the previous posterior $p(S_1, S_2|t_1, t_2)$ with two terms; the likelihood $p(t_3|S_1, S_2, S_3, t_1, t_2)$ and a prior for how likely the next 'branch' is, $p(S_3|S_1, S_2)$., which is the CRP presented above.

We now consider how to determine the likelihood and prior probabilities. The first source can only be associated with one source, so $p(S_1 = 1) = 1$. The principle of Occam's razor would suggest that $p(S_1 = 1, S_2 = 1) > p(S_1 = 1, S_2 = 2)$, i.e. if we haven't heard any of the sounds, the most probable acoustic scene is the simplest one: all sounds come from the same source. The value of $p(S_1 = 1, S_2 = 1)$ for an individual can be determined from fitting their data, and the value $p(S_1 = 1, S_2 = 2)$ is simply $1 - p(S_1 = 1, S_2 = 1)$. Note also that due to Eq. 3, and that by definition $p(S_1 = 1) = 1$, we can write $p(S_1 = 1, S_2 = 2) = p(S_2 = 2|S_1 = 1)p(S_1 = 1) = \alpha/(1 + \alpha)$. The alpha value may depend on factors such as the environment, which are not considered in the model: natural signal statistics may provide guidance for how the prior probabilities are assigned.

Regarding the likelihood function, for new tone $i$ the observer assumes the generative probability $p(t_i|S_i, t_{i-k}, S_i = S_{i-k})$, where tone $i - k$ was the latest tone inside stream $S_i$ (see Equation 1 above). Note that this applies even when the sounds generated by the same source are separated by one or more sounds associated with different sources (e.g., $(S_1 = 1, S_2 = 2, S_3 = 1)$). The only transition that matters is that between the most recent tone and the last tone in the same stream, so if three tones $t_1$, $t_2$ and $t_3$ had all been associated with the same stream (e.g. $(S_1 = 1, S_2 = 1, S_3 = 1)$), we would only consider the transition from $t_2$ to $t_3$, whereas if $t_2$ was associated with a different stream (e.g. $(S_1 = 1, S_2 = 2, S_3 = 1)$), we would only consider the transition from $t_1$ to $t_3$.

If a sound comes from a new source, then we assume that the likelihood is independent of previous tones:

$$p(t_i|S_1 \neq S_i, S_2 \neq S_i, ...S_{i-1} \neq S_i) = \frac{1}{\sqrt{2\pi\theta^2}}exp^{-\frac{(f_i - f_0)^2}{2\theta^2}}$$

(6)

where $f_0$ is the midpoint of the range of auditory tones presented for the trial (the model is insensitive as to whether the mean is chosen instead). Prior to being presented with any

tones it is reasonable to expect that the first tone will be within the range of tones previously experienced in the experiment. For the variance of this likelihood, $\theta$, we have set it to be an arbitrarily large value, to reduce the number of variables in the model and as it plays a limited role in practice.

Fig 2 summarizes the main properties of the model. The final model has two parameters, $\alpha$ and $\sigma$, plus a parameter for the steepness of the response variability (given by softmax function) for each subject $\beta$. For details of implementation of the model, see the Methods section.

## Results

To evaluate the performance of the model, we made qualitative comparisons with studies in the literature and quantitative comparison with experimental data. We furthermore tested a prediction from the model using a novel experimental paradigm.

### Modeling

**Time and frequency.** A well-known basic stream segregation phenomena (e.g. [11]) shows that increasing the speed at which auditory tones are presented, increases the probability that tones are perceived as coming from separate streams. To examine this in the model we recreate the second experiment of Bregman and Campbell [11], showing in Fig 3a–3b that while a sequence of six slowly presented tones is assigned a low posterior probability of originating from different sources (and should therefore be assigned the same stream), as the speed of presentation is increased the probability of originating from two sources increases drastically (implying subjects should segregate the streams). Likewise, separation by frequency causes the clustering of tones to be more difficult. Van Norden [1] found that in a repeating low-high-low sequence of tones (see Fig 3), subjects would report one or two streams as a function of both the difference in auditory frequency and the speed of presentation (inter-stimulus interval). In our computational model (see Fig 2) the likelihood term directly depends on both of these factors, and the prior probability restricts the observer from segregating tones into more streams. This galloping effect is replicated by the model (Fig 3c–3d) which assigns a high probability to the lower and higher tones coming from separate sources, and thus assigns them to separate streams. We also replicate the results of this study in Experiment 1 below.

**Cumulative.** The galloping sequence Low-High-Low has also been used to highlight the effect of the accumulation of information (Fig 4). Bregman [26] showed that a short sequence of tones tends to lead to the percept of a single stream, whereas the accumulation of information causes the tones to segregate into two streams. For the model, this effect is due to the non-parametric prior initially assigning a low probability to the possibility of two streams, before more information is gathered.

By default, the model is biased towards the most parsimonious interpretation of events (Occam's razor), which in this case means creating as few different causes as possible. This is a direct consequence of the rich-gets-richer property of the CRP, because the model needs to accumulate a sufficient amount of evidence before deciding to abandon the initial single stream hypothesis.

**Context.** An aspect of auditory perception that especially received attention from the Gestalt movement was the role of context in auditory clustering. Experiments done by [27] showed that modifying the context in which tones were presented modified the segregation of unmodified tones. Fig 4 shows an example, based on [27], where two low tones will be

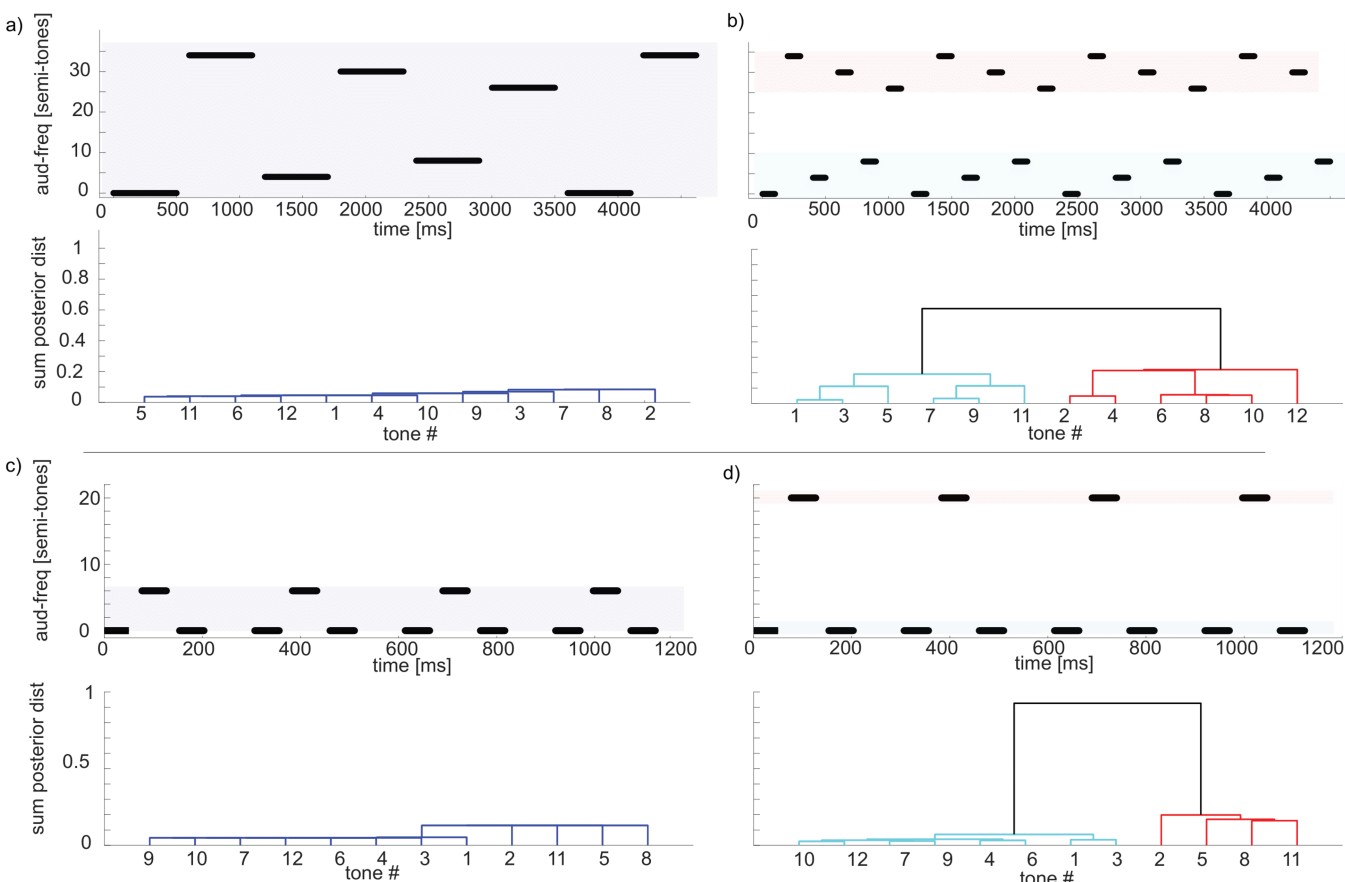

**Fig 3. a–d) Stimuli used in experiments from [11] (second experiment), highlighting how the speed of presentation affects perception of streams of tones.** Stimuli are shown at the top, bottom are dendrogram tree-plots based on the posterior distribution over clustering. A unique colour is assigned to clusters with more than 50 percent distance from other clusters. a) Slow sequence, ISI 100 ms, tone duration 500 ms, pitch difference [0 4 8 26 30 34 ] semi-tones, tone sequence repeated twice. The posterior mode (the sequence combination with the highest posterior probability) was 111111, i.e. all tones assigned to the same stream. b) Fast sequence, ISI 100ms, tone duration 100 ms (posterior mode 121212). c-d) Example of a galloping stream, from [1], highlighting effect of frequency differences. c) ISI 26.6ms, pitch difference 6 semi-tones (posterior mode 111) d) ISI 26.6ms, pitch difference 20 semi-tones (posterior mode 121). Parameters for this figure (and subsequent figures) were $\alpha = 1.44$, $\sigma = 40$.

clustered together while two distractor tones are far off in frequency, but will be clustered separately as the distractor tones are placed around them. The model replicates this phenomenon, showing a separation of the first two tones in Fig 4c–4d.

In this scenario, the low tones from Fig 4d are alternating fast enough to be segregated into two different streams despite the fact that their frequencies are close. However, when alternating high tones are added, the implementation of Occam's razor through the CRP prevents the multiplication of streams: the frequencies of the low and high tones are far enough to segregate into two streams, but as the probability of creating a new stream decreases every time one is created, they are now too close within those streams to warrant the creation of a third and fourth streams.

**Crossing.** As shown by Tougas and Bregman [28] interleaving a decreasing and increasing series of tones gives the illusory percept of the two streams 'bouncing' i.e. the lower set of tones is clustered and segregated from the higher set of tones. Fig 5 recreates this experiment with $2 \times 10$ interleaved decreasing and increasing tones. The model recreates the perceptual

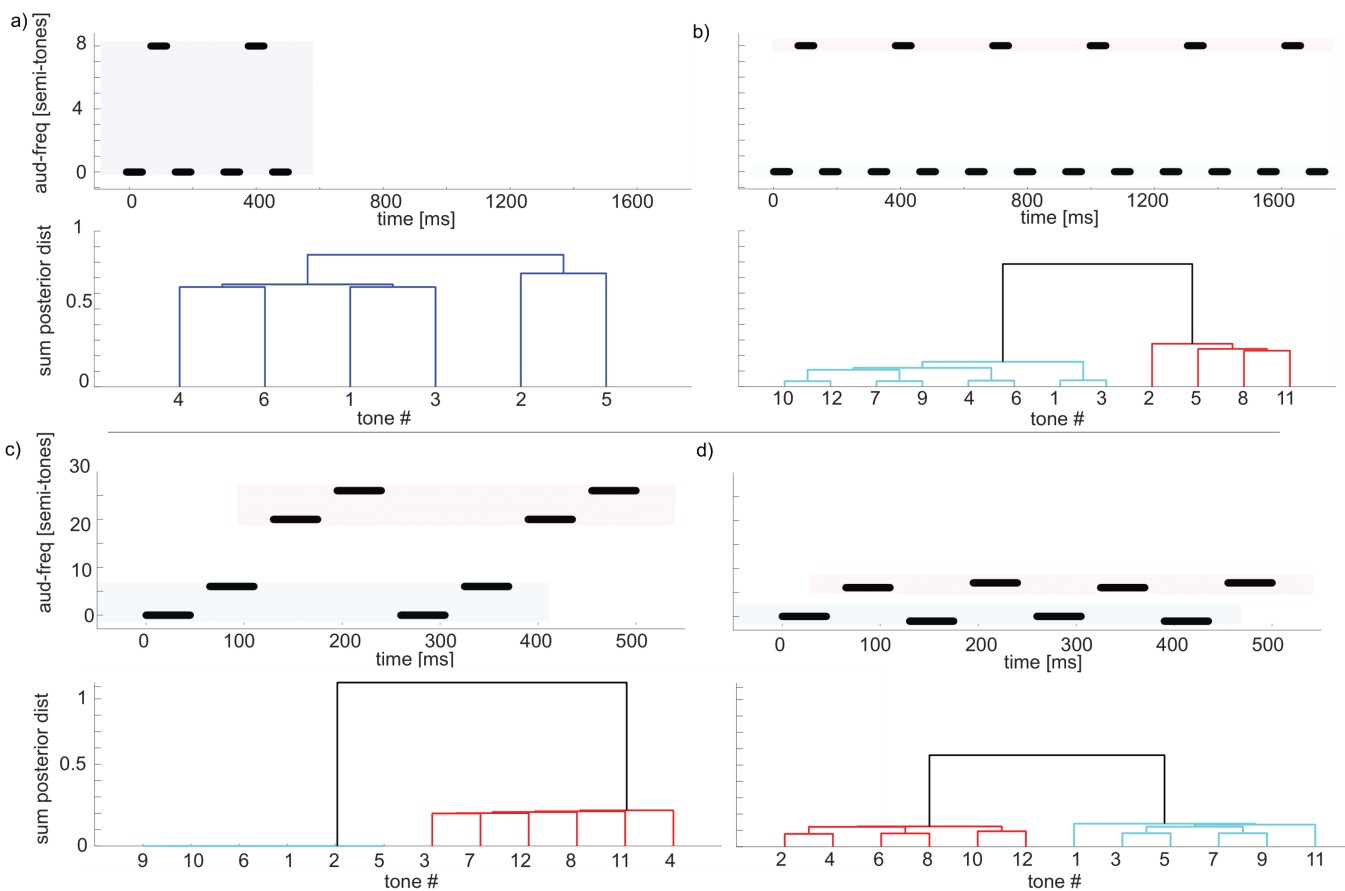

**Fig 4. a–b) Stimuli used in experiments from Bregman [26], highlighting the cumulative effect of tones.** Stimuli are shown at the top, and at the bottom are dendrogram tree-plots based on the posterior distribution over clustering. A unique colour is assigned to clusters with more than 50 percent distance from other clusters. a) Short sequence ISI 26.6ms, pitch difference 7 semi-tones, tone sequence repeated twice (posterior mode 111). b) Long sequence ISI 26.6ms, pitch difference 7 semi-tones, tone sequence repeated eight times (posterior mode 121). c-d) Context matters for the clustering of tones. c) Two low tones , two high tones, leading to low tones segregated from high tones (posterior mode 1122); d) While the two low tones have been kept constant, the context of the two other tones now causes them to be clustered separately with the other tones (posterior mode 1212). Long sequence ISI 26.6ms, tone sequence repeated eight times. The modeling parameters were the same as in Fig 3.

phenomenon, with the lower frequency tones grouped together, separate from the higher frequency tones, thus implying a perceived 'bounce'.

The model has ample evidence at the beginning of the sequence to infer the existence of two different streams. Once tones seem to cross, the model is not trying to predict the frequency of the next tone in each stream through their trajectory (there is no concept of momentum in the model); instead, every new tone is compared to the latest tone of each stream, and the stream that minimizes $\Delta f/\Delta t$ will be selected. In the current situation, this makes low tones and high tones stay segregated regardless of their trajectory.

**Comparison to other models predicting behavioural phenomena.** Overall, the described model is able to recreate several phenomena in the experimental literature. While there have been previous studies modeling auditory scene analysis, most have focused on single, or few phenomena [15,29–31].

A previous study provided a mechanistic model [32] which is able to recreate a similar number of phenomena (implemented differently in [33]) and make qualitative predictions

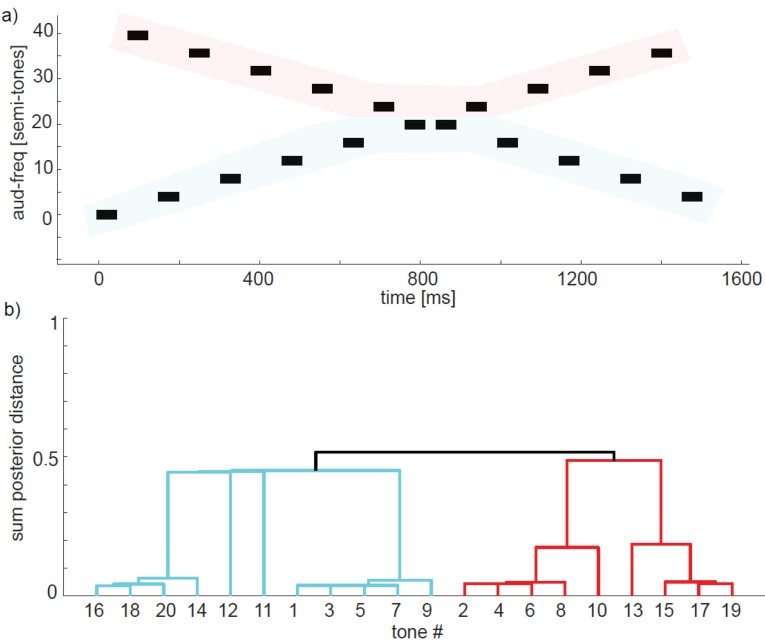

**Fig 5. Interleaved increasing (uneven numbered tones) and decreasing (even numbered tones) series of tones, ISI 26.6ms.** Same as for human observer the model assigns higher value to a 'bouncing' percept, where tones [2 4 6 8 10] are clustered together with [13 15 17 19]. Modeling parameters were the same as in Fig 3

for behaviour (note the similarity between Fig 6 below and figure 6 in [32]). Their model is able to replicate effects on stream segregation of speed of presentation (Fig 3a above), auditory frequency (Fig 3b), multi-tone cycles (context, Fig 4b), and crossing (Fig 5), as well as effects using complex tones which we do not address in this paper. However, this model is not able to explain the build-up effect (Cumulative, Fig 4a above) that happens initially when an ambiguous 1-stream percept becomes a 2-stream percept after increased exposure within a trial. In order to explain such an effect the model by Elhilali et al. [32]) would need to add a biological adaptation, through e.g. synaptic depression.

The study by [34] also used a mechanistic model to replicate the effect of speed, audio frequency, and cross-over. They did not replicate the build-up effect, or examine effect of context, but did show a novel effect of amplitude.

Neither of these studies performed any quantitative model comparison to behavioural data (for a discussion of the principled differences between the models, see Discussion). An advantage of our approach is that our quantitative model allows us to perform model fitting and model comparison, to assess the importance of different aspects of the model, as we will do for the galloping effect below.

## Experiment 1

To quantitatively compare the model with human performance, we conducted a psychophysics experiment, in which 15 participants with normal hearing listened to simple auditory sequences and performed a subjective judgement task (a variant of the galloping experiment [1], see Fig 3). Given a sequence of Low-High-Low tones, subjects would respond whether they perceived one or two streams. Across trials, the separation between low and high tones, and the inter-stimulus-interval, were varied (see Methods for details).

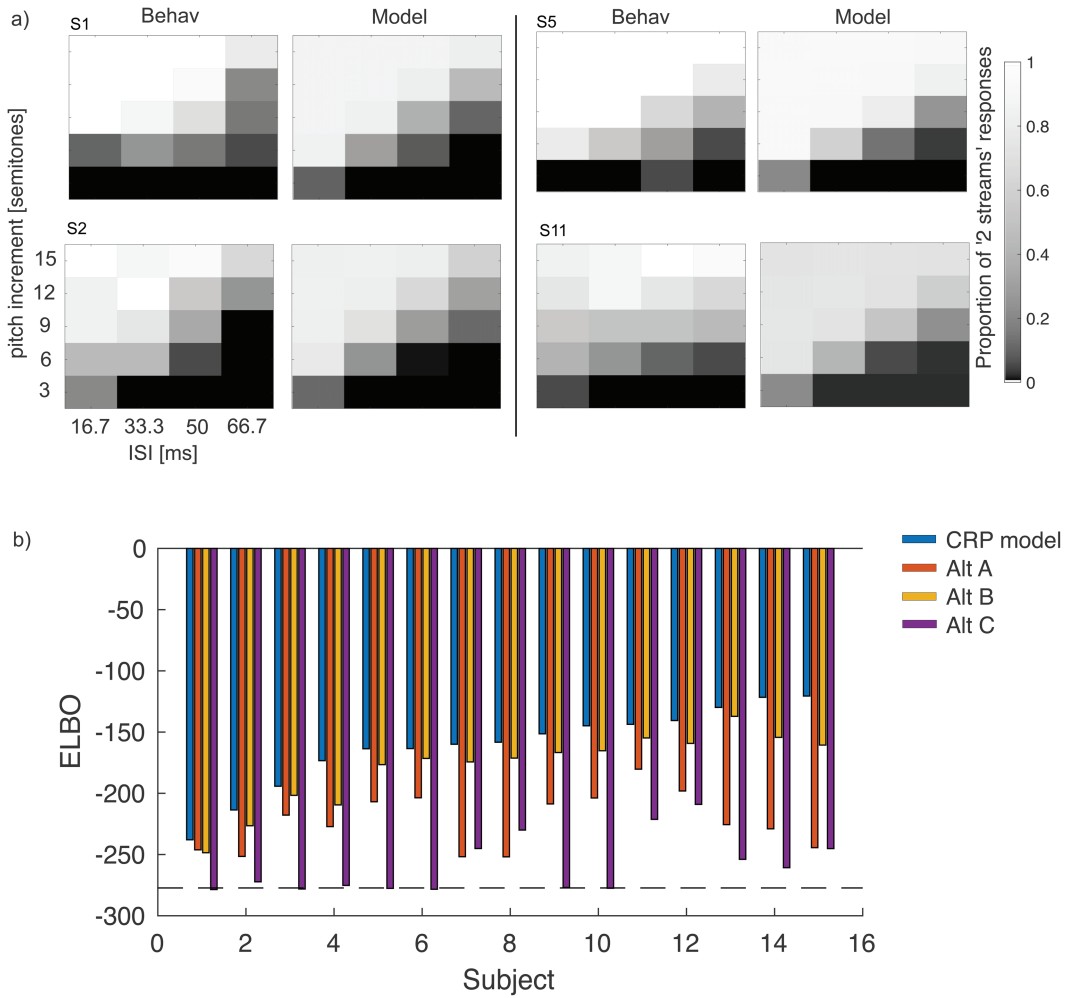

**Fig 6. a) Behavioural data and model simulations after fitting for four subjects, giving the fraction of trials in which the participant responded '2' for the number of streams perceived.** Axes give the pitch difference for the middle tone and the inter stimulus interval (ISI): the time between the offset of one tone and the onset of the next. b) Model performance on experiment 1 in terms of Evidence Lower Bound (ELBO) for each subject with the CRP model (dark blue), alternative A (red), alternative B (yellow), and alternative C (purple). The black dotted line indicates the performance of a purely random model that assigns 0.5 probability to either response for every condition. Subjects are ordered based on CRP model ELBO values. In order to find a measure of the overall performance of the CRP model we calculated the average relative ELBO between random and perfect model fit (ELBOmin-ELBO)/ELBOmin). This average ELBO proportion is 0.419 for the CRP model, implying a good fit.

**Model performance and comparison.** As an example, response data from four subjects is shown in Fig 6b. As expected, when the ISI was short or when the difference in frequency was large, subjects were more likely to report two streams than one. Fig 6 also shows the fit of the model for the same four participants (see Fig B in S1 Text for all subjects and model fits).

While visually the model approximates the subject behaviour, we used model-comparison to rule out other hypotheses.

The model with the non-parametric prior was compared against three alternatives that used different priors to constrain the number of possible streams:

A. When the stream combination comprised only one stream (repeated), the prior probability of the next stream being 1 or 2 was allocated according to the CRP, but if the combination already contained two streams, the prior probability of allocating stream 1 or 2 was simply the fraction of previous tones that were allocated to stream 1 or 2 respectively.

B. The prior probabilities of a new tone being allocated to stream 1 or stream 2 were given by P1 and 1-P1, respectively, where P1 is a fixed parameter.

C. Prior probabilities that a new tone was assigned to stream 1 or 2 were fixed at 0.5.

For model comparison (and fitting) we used the Evidence Lower Bound, as implemented in the VBMC package [35]. The Bayesian model with the non-parametric process prior gives a better fit (larger ELBO) than all the alternatives considered for every subject, with a protected exceedance probability [36] of $p < 0.001$ for the other three models. The mean $\pm$ SEM of the optimised parameters for the unconstrained model are $\alpha = 4.68 \pm 0.23$ (equivalently $P(11) = 0.184 \pm 0.010$) and $\sigma = 107.4 \pm 6.8$ [semitones/sec].

## Experiment 2

While the model above theoretically allows an unlimited number of tones to be segregated into an unrestricted number of streams, Experiment 1 (presented above) only allows a repeated sequence of 3 tones to be separated into 1 or 2 streams. However, both the model and the Auditory Scene Analysis literature predict that subjects should generally segregate based on the frequency and temporal distance between tones with a possibly infinite number of streams. To test this further, we performed a novel follow-up experiment in a broader auditory environment. The experimental setting was inspired by Barsz [37] and was specifically designed to allow three streams to emerge and to replace the explicit measure of stream segregation by an implicit one (see Methods for details).

Participants were asked to judge if two consecutive melodies comprised of 4 repeated tones were similar or different. In every condition, 1 tone was considered to be in the low frequency range, 2 tones in the medium range, and 1 tone in the high range. According to previous experiments showing that order information is intact when tones are inside the same stream but lost when segregated into different streams [11,37,38], if medium tones were to be part of a stream excluding low and high tones, participants should be unable to detect the difference between two sequences if only medium tones were inverted. Indeed, the stimulus onset asynchrony between medium tones remains constant within the melody despite the inversion; furthermore, each melody progressively faded in and out in order to mitigate the participants' ability to infer a melody difference by simply noticing which medium tone came first or last. Thus, subjects should only be able to detect the inversion of the middle tones if the tones are placed in the same stream as the upper or lower stream (hence all tones clustered into 1 or 2 groups). Conditions were created with varying discrepancies between the low and medium frequency ranges, and the medium and high frequency ranges, counted in semitones (ISIs were kept constant for this experiment). For instance, the *3-9* condition means that the lowest medium tone is three semitones higher than the low tone, and the highest medium tone is nine semitones lower than the high tone. See Methods for a schematic representation of a typical stimulus with inversion.

As our model predicts that the probability of being assigned to different streams is dependent on the frequency difference, our first prediction is that participants should have a reasonable degree of performance in detecting a difference between two melodies when the medium tones are inverted and there is only a small frequency difference between those

medium tones and at least one of the high and low tones. This would reflect the fact that medium tones are in a stream that also includes other tones. Conversely, our other prediction is that as the frequency difference increases, participants should perform significantly worse at detecting the medium tone inversion. This would reflect the fact that medium tones are in a stream that does not include other tones. This is in contrast to any model that only allows two streams (such as models B and C above), according to which large frequency difference should not lead to any decrement in performance.

**Analysis preparation.** Individual responses for perceived differences between sequences were transformed into D-prime scores ($d' = Z(HR) – Z(FA)$ where $Z$ is the inverse of the normal cumulative distribution function, HR is the hit rate and FA the false alarm rate) to obtain a single measure of signal detection for each pair of frequency differences, while taking into account possible response biases. Hence a condition where all tones are clustered together would make the task easy, and give a high hit rate and low false alarm rate, and thus a high D-prime. In contrast, when the middle tones are independent from the high and low tones the D-prime should be low. Two participants with a negative D-prime score in the easiest condition (*3-3*) were considered unable to perform the task correctly and were therefore excluded from further analysis, leaving a total of 24 participants. No data was missing from the data set. A one-way repeated measures ANOVA was conducted on these D-prime scores, with FREQUENCY DIFFERENCES (*3-3* vs. *3-9* vs. *3-15* vs. *9-9* vs. *9-15* vs. *15-15*) as the only within-subject factor, along with seven subsequent paired-sample t-tests in line with our hypotheses.

**Data analysis.** Although inferential statistical tests were conducted only on D-prime scores, Table 1 also includes the summary statistics of the proportions of "similar" responses in every experimental condition.

One-way repeated measures ANOVA (FREQUENCY DIFFERENCES) showed that D-prime scores differed as a function of this factor [$F(5,115) = 6.659$, $p < .001$]. This is in accordance with our CRP model, but against predictions from any model that only allows 2 potential streams.

Seven subsequent paired-sample t-tests were performed to decompose the main effect of FREQUENCY DIFFERENCES over D-prime scores. Three paired-sample t-tests revealed that D-prime scores were significantly higher in the *3–3* condition than in the *9–9* condition [$t(23) = 3.003$, $p = .006$, $d = 0.613$], in the *9–15* condition [$t(23) = 2.556$, $p = .018$, $d = 0.522$] and in the *15–15* condition [$t(23) = 2.535$, $p = 0.019$, $d = 0.517$]. Two other paired-samples t-tests revealed no significant difference between the *3–3* and the *3–9* conditions [$t(23) = 0.681$, $p = .503$, $d = 0.139$] and between the *3–3* and the *3–15* conditions [$t(23) = –1.568$, $p = .131$, $d = –0.32$]. Another two paired-samples t-tests revealed no significant differences between the *9–9* and the *9–15* conditions [$t(23) = –1.02$, $p = .318$, $d = –0.208$] and between the *9–9* and the *15–15* conditions [$t(23) = –0.949$, $p = .353$, $d = –0.194$]. These results are summarized in Fig 7.

**Model performance.** While the results above support the ability of the model to explain to qualitatively predict the performance of subjects on a novel experiment, we also wanted to verify this quantitatively. We fitted the model (see Methods) to the data for each subject, with the same model variants as for Experiment 1. We again found that the model with a

**Table 1. Mean proportions of hits and false alarms of "similar" responses, and mean D-prime scores across all conditions. Reported errors are ± 1 standard deviation.**

|                 | 3-3           | 3-9           | 3-15          | 9-9           | 9-15          | 15-15         |
|-----------------|---------------|---------------|---------------|---------------|---------------|---------------|
| False-alarm rate | 0.271 ± 0.209 | 0.288 ± 0.292 | 0.240 ± 0.258 | 0.486 ± 0.253 | 0.497 ± 0.284 | 0.538 ± 0.317 |
| Hit rate        | 0.799 ± 0.230 | 0.771 ± 0.240 | 0.833 ± 0.197 | 0.771 ± 0.202 | 0.854 ± 0.199 | 0.896 ± 0.176 |
| D-prime         | 1.476 ± 0.765 | 1.371 ± 1.016 | 1.713 ± 0.863 | 0.748 ± 0.843 | 0.976 ± 0.777 | 0.960 ± 0.940 |

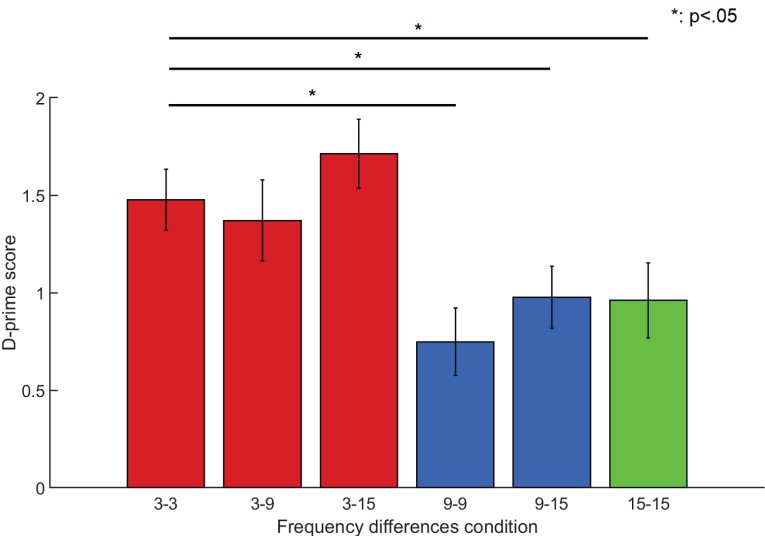

**Fig 7. D-prime scores as a function of frequency difference. Red bars indicate conditions with a small minimum frequency difference, blue bars indicate conditions with an intermediate minimum frequency difference and green bars indicate conditions with a large minimum frequency difference.** Error bars are ± 1 standard error

CRP prior was best able to explain the data with an average advantage of ELBO for the CRP of (-1.9204 -3.6681 -1.5003) relative to the other three models. Furthermore, the protected exceedance probability [36] was respectively ( 0.9993 0.0003 0.0002 0.0003) indicating a strong preference for the CRP model (see Fig 8).

It is worth noting that the behavior of the subjects on this task is more noisy and even the best-performing model is at chance level for 6 of the subjects.

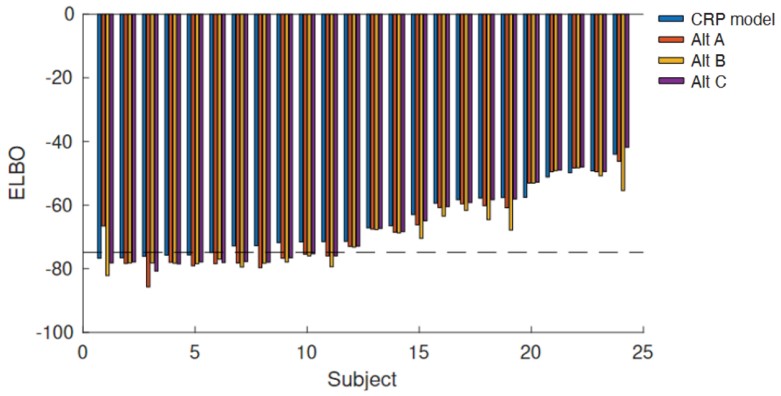

**Fig 8. Model performance on experiment 2 in terms of Evidence Lower Bound (ELBO) for each subject with the CRP model (blue), alternative A (red), alternative B (yellow), and alternative C (purple).** The black horizontal dotted line indicates the performance of a purely random model that assigns 0.5 probability to either response for every condition. Subjects are ordered based on CRP model ELBO values. Large negative values indicate poor performance of a model. The average ELBO proportion (calculated as in Experiment 1, where 0 is random and 1 is perfect fit) was 0.127.

## Discussion

We have presented a simple Bayesian perceptual model that is capable of assigning stimuli to an unrestricted number of sources, by clustering of stimuli. We have applied the model to the specific case of auditory stream segregation, an area where Gestalt psychology has long emphasized the need for grouping. Utilizing a non-parametric Bayesian prior, the model iteratively updates the posterior distribution over the assigned group of each stimulus and provides an excellent description of the perceptual interpretation of simple auditory sequences by human observers.

With just two parameters, the model gives a good account of the basic characteristics of auditory stream segregation – the variation in the probability of perceiving a single sound source as a function of the repetition rate and pitch difference of the sounds. The basic model (with a softmax decision function) gave a better fit to experimental data (Exp. 1) than alternative models that were constrained to interpret the sounds as being produced from just one or two streams. Predictions from the model were also in accordance with results from a novel experiment with a larger number of tones (Exp. 2).

Importantly, the model goes beyond giving just the number of sources, but says which sounds are produced by each source. While the combinatorial space of the posterior distribution in experiment 1 was collapsed to give a marginal distribution in a continuous 1-d response space (leading to an estimate of response probability), the maximum a posteriori (MAP) for all participants was always located at either 111-111... or 121-121..., depending on the stimulus condition (Fig 3). This is reassuring as it is consistent with the anecdotal evidence that participants always perceive either a galloping rhythm (streams 111-111...) or a high-pitch and a low pitch stream (121-121...), i.e. the percept is always at the MAP. Indeed, the percept cannot in general be at the mean because the space of possible percepts is discrete: there is no percept in-between 111 and 121. Specifically, to respond in our experiments, alternative strategies may be employed, such as posterior sampling [39], which may be challenging to distinguish from our current methodology, or heuristic strategies. More advanced experimental methods and modeling would have to be used to separate these alternatives.

One consequence of the inference model that is not addressed by mechanistic models of stream segregation is that when a percept changes from, say 111-111 to 121-121, the source allocation of previous sounds is changed. Ironically, this 'non-causal' effect is essentially a feature of causal inference – when an observer decides that the percept has changed to 121-121, this is based on previous evidence, and yet at the time that the previous tones were heard, they were all associated with one source. A similar effect is commonly encountered when mis-interpreted speech (perhaps mis-heard due to background noise) suddenly makes sense when an essential word is heard – the previous words are reinterpreted, similar to the letters in predictive text message systems. Recent work has seen increased interest in examples of predictive postdiction [40], i.e., phenomena in which the perception of a set of stimuli is influenced by stimuli appearing later, seemingly without a causal link [41]. However, as we have highlighted in this paper, a causal interpretation based on inference over the full set of stimuli provides a normative explanation.

While the current implementation improves on the intractability of traditional parametric models when the number of possible sources exceeds a few, numerical approximations were still necessary to manage the model's complexity. This brings the model closer to mimicking the brain's constant, automatic, and effortless causal inference. Due to the potential complexity of doing inference over a large number of stimuli, and the exploding number of possible interpretations, any computational system will need to employ approximations or

simplifications. What approximations or simplifications the nervous system uses is as of yet unclear.

As an alternative to calculating our results analytically (but with approximations), we could use Monte Carlo techniques (e.g. Markov Chain Monte Carlo sampling, i.e. MCMC, a different type of approximation), which have become a standard tool for solving complex statistical models. While not presented here, we have also implemented a MCMC version of the model with similar results. In future work this might be useful in explaining bistable behavioral phenomena, where an observer will seemingly randomly switch between incongruous interpretations of stimuli [42,43], a phenomenon not explained by the current model.

The framework of the model is very general and allows for the incorporation of other factors into the likelihood to describe other aspects of auditory stream segregation. Adding terms in the likelihood function may be able to explain other effects seen in the literature, such as segregation based on bandwidth [44], or build-up and resetting of segregation [45]. Furthermore, in the current study we assume that there is no ambiguity in the percept of the pure tones, the uncertainty arises from the lack of knowledge about the underlying generative structure of the data. In a realistic situation perceptual ambiguity would have to be taken into account using an approach such as suggested by Turner and Sahani [46]. Nevertheless, we should emphasize that even though we are dealing with a Markov property (each tone within a stream only depends on the previous tone within the stream), the mixture of streams makes the problem very different from the work on e.g. Hidden Markov Models (or even Infinite Hidden Markov Models) for which the goal is to infer the underlying states despite noisy signal [47].

Our approach is an example of a predictive coding model [48] that uses a generative model to make predictions about the relationships between stimuli and subsequently update these relationships. The model shares similarities with work by Barniv and Neike [42], who used Gaussian mixture models to describe the integration and segregation of auditory objects, and assumed that a new object could be introduced with a small fixed probability. The model implemented by Chambers et al. [30] also used a similar probabilistic framework and Gaussian likelihood to explain the perceived continuity of pure tones with octave relationships ('Shepard tones'). In contrast to such probabilistic models, there have also been proposals for mechanistic models, such as neural network models simulating the abstract processing of the cochlea and auditory cortex [33], as well as models explaining behavioural phenomena through interactions between different neural populations [15,29].

The proposed model of auditory stream segregation is a specific instantiation of an iterative probabilistic approach towards inference of perceptual information. A major issue for this approach is the problem of dealing with multiple sources, as represented by the work done on causal inference [2,3,49]. Until now, models of causal inference have been unable to handle more than two sources, due to the increasing number of parameters needed for parametric priors. The use of a non-parametric prior allows a complex of many stimuli to be interpreted without running into this problem, potentially allowing for an arbitrary number of causes in the world. This approach is very general - it can be applied to any set of discrete sequential cues involving multiple sources - and it gives a simple, principled way to incorporate signal constraints into the generative model. In the current study we do not assume any perceptual noise or variability, instead assuming variability in how pure tones change over time (the generative structure). In contrast, work on perceptual causal inference has assumed perceptual noise [2,49]. Future work will have to combine the non-parametric models with perceptual causal inference, including perceptual noise.

While frequency and time are the most studied dimensions in the Auditory Scene Analysis literature [9,12], the general idea of explaining perceptual clustering through the use of a

Bayesian non-parametric prior can potentially be applied to other auditory variables, such as harmonic complex tones [50] or multiple speakers, i.e. the cocktail party problem [51]. It is also possible to add other features in addition to time and frequency, to the existing model, such as spatial discrepancy and saliency, topics we plan to explore in the future.

The auditory streaming model currently also does not consider attentional mechanisms. As presented here, it assumes that our perceptual system automatically groups or segregates cues into a potentially infinite number of streams based on stimulus characteristics, and that attention only has the role of selecting one or several clusters of interest in a given situation, or is being driven by the posterior probability of the different clusters. In either case, attentional processes would be entirely independent of the mechanisms simulated by the model. Nevertheless, it has already been proposed that attention could interact with the cluster formation process itself and therefore affect their overall configuration [52]. If that is the case, the model's responses could significantly differ from behavioural results in situations where top-down attention is being manipulated.

Auditory stream segregation is considered a typical example of the Gestalt principle of Pragnanz [4], according to which properties of a set of stimuli are interpreted in a manner that simplifies the interpretation. We would propose that another way to interpret this is by the principle of clustering, with stimuli that are similar (in auditory frequency, visual continuity etc.) being combined into a single simple interpretation. The simplification principle is naturally interpreted in our framework through a non-parametric Bayesian process that naturally restricts the complexity of the interpretation, such as the number of possible objects or sources in a scene. Recent work has also shown how visual motion can be modeled effectively through a clustering perspective [53].

Ultimately, the processing of auditory stimuli in an organism must be for the goal of survival and procreation. Simplicity, as encapsulated in this model, is not a goal in itself but can be thought of as a heuristic towards improved detection, tracking and processing of auditory sources. Further work in this area can expand on under what conditions this is indeed a useful heuristic.

In conclusion, we have shown that auditory scene perception of streams of single frequency tones can be explained by a simple Bayesian model utilizing a non-parametric prior. This highlights the importance of clustering in auditory perception, although the approach is applicable to any combination of stimuli and perceptual cues. Together with advances in visual perception [7] this hints at clustering being a general property of perception.

## Materials and methods

### Ethics statement

The consent of the subjects were obtained in writing and the two experiments were approved by University of Birmingham and Durham University's Ethics Committees respectively (refs. ERN11-1072 and 15/55).

### Modelling

**Model response and fitting, Exp 1.** For experiment 1, to determine the response (1 or 2 streams) of the model to a tone sequence, the posterior for each possible sequence, $S_{1:n}$, is calculated tone-by-tone until all 30 tones (maximum) have been presented. To relate the final posterior over sequences to response $r_k$ of subject $k$, ('1 or 2 streams') $P_{model}(r_k|tones)$, we

assume that subjects maximise the expected utility

$$_{r_k} < U >=_{r_k} \sum_{i,j} U(r_k, S_i, S_j) * P(S_i = S_j | f, t) \tag{7}$$

where the utility of a response given two tones is counted as 1 if they are in the same stream, and 0 otherwise. Note that the absolute values of $S_i$ do not matter, just whether they are in the same stream or not.

$$U(r_k, S_i, S_j) = 1 \quad \text{if } (r_k = 1, S_i = S_j) \text{ or } (r_k = 2, S_i \neq S_j) \tag{8}$$
$$U(r_k, S_i, S_j) = 0 \quad \text{if } (r_k = 2, S_i = S_j) \text{ or } (r_k = 1, S_i \neq S_j) \tag{9}$$

The best response is then to choose 1 (single stream) if

$$\sum_{i,j} P(S_i = S_j | f, t) > \sum_{i,j} (1 - P(S_i = S_j | f, t)) \tag{10}$$

If the observer believes all tones came from the same stream they should choose $r_k = 1$, and $r_k = 2$ otherwise. The model can freely explore hypotheses related to more than two streams, but will answer '1 stream' or '2 streams' to reflect the forced response choice from participants.

We assume soft-max, a variant of probability matching similar to Luce's law [54], to explain the variability in data and allow us to fit our models.

$$P(r_k | f, t) = \frac{exp(\beta * \sum P(S_i = S_j | f, t))}{exp(\beta * \sum P(S_i = S_j | f, t)) + exp(\beta * \sum (1 - P(S_i = S_j | f, t)))} \tag{11}$$

Our standard CRP-based model had two parameters, $\alpha$, which specifies how fast clusters in data form, and $\sigma$ which specifies how sensitive the participant is to changes in frequency. A steepness parameter $\beta$ was also used purely for fitting the model. The parameters of the model (as well as for the alternative models) were optimised independently for each subject $k$ using the BADS and VBMC toolboxes (as a more robust alternative to MATLAB's fminsearch routine) [35,55,56], to maximise the Evidence Lower Bound (ELBO) based on the likelihood $\log P_{model}(r_k | tones, par)$ combined with a weak prior. During each iteration of the search, a sequence of 30 tones was presented to the model for each condition, and the probability of response '1' was calculated per condition.

We performed six initialisations of the VBMC algorithm for each subject (to avoid fitting to a local minimum), and assessed the convergence using the built-in diagnostic function. When reporting values from VBMC (parameters or ELBO) we always report the mean values.

**Model posterior approximation.** Using the iterative scheme above, we can calculate analytically the possible combinations of tones, but as the tone sequence progresses the number of possible source combinations - and hence the size of the posterior distribution - increases exponentially. To prevent combinatorial explosion two methods were used to generate an approximation of the full posterior distribution. The first limits the number of tones that are retained when using the previous posterior as the next prior, i.e. the algorithm only retains the last 30 tones and their potential allocations to sources

Limiting the number of tones eases the computational load and can also be seen as a crude model of a limited memory capacity. Although the iteratively constructed prior retains some stream information of all previous tones, when a very short memory is used, this may not be

sufficient to generate stable stream allocation as the CRP prior probabilities fluctuate greatly when the number of previous tones is small. Furthermore, if the structure of the sequence is an important cue for streaming, a larger memory may be necessary to determine regularities in the sequence.

Even when the memory is limited to (e.g.) the previous six tones, allocating a stream to the seventh tone requires a posterior distribution taking 858 values, most of which must necessarily have very small probabilities. A second method to limit the size of the posterior is simply to select only the most probable stream combinations by imposing a probability threshold, hence we only propagated stream combinations with $P(S_{1:n}|tones_{1:n}) > 0.001$. Together, these approximation methods allow a reasonable memory length of 30 tones (to avoid instability), while avoiding combinatorial explosion.

We performed a parameter recovery, as well as model recovery exercise, for the model in Experiment 1 (see sections Parameter recovery, and Model recovery in S1 Text.

**Model response and fitting, Exp 2.**   The qualitative and quantitative predictions for Experiment 2 rest on the assumptions that subjects will be unable to differentiate the order of the middle tones (M1 and M2) if the tones are in a stream only with themselves, i.e. if L and H are in separate streams from M1 and M2. Similar to methods for Exp 1 we marginalise over all the probabilities where

$$P\big(S_{M1} = S_L \& S_{M1} = S_H \& S_{M2} = S_L \& S_{M2} = S_H | f, t\big) = 1 - \sum_{n,o} P\big(S_n = S_o\big)$$

where $n \in (M1, M2)$ and $o \in (L, H)$.

Same as for Experiment 1 we assumed a soft-max distribution of responses to explain variability of responses and help us fit our models.

In contrast to Experiment 1 we did not fit the prior probability of a single stream to the data ($P(11) = 1/(\alpha + 1)$), but instead assumed a fixed value across subjects of 0.18 (mean fitted value from Exp 1). The reason for this was that while in Exp 1 we varied both the tone differences and the tone timing gap, allowing us to examine effect of parameters for prior probability $P(11)$ and the variance $\sigma$, in Exp 2 we only varied the tone differences thus giving us less dimensions of variability and thus need to restrict the number of parameters.

Otherwise the fitting worked in a similar manner using the VBMC toolbox to find the distribution of ELBO.

We performed a parameter recovery exercise for the model in Experiment 2 (see section Parameter recovery in S1 Text).

## Experimental setup

Two experiments were conducted to test different aspects of the model. Experiment 1 was a replica of the 'galloping' stimuli experiment [1] performed to test the quantitative performance of the model, while the second experiment was a novel task designed to test one of the predictions of the model.

Subjects for both experiments were under-graduate students and received course credits for their participation, except for one of the authors who participated in Experiment 1. Each subject was fully briefed, provided informed consent and was given brief training on the task they performed. Experiments were completed by different subjects. No personal data was kept, ensuring participants' anonymity.

Stimuli were dynamically programmed using Matlab [57] on a PC desktop computer. Both experiments used the Psychtoolbox extension to ensure timings were accurate [58].

Stimuli were played through Sennheiser 280 headphones at a comfortable supra-threshold level plugged into an external sound card. The experiment was carried out in a special sound-attenuated room.

**Experiment 1.** This experiment replicated the study from [1], using fifteen participants. Fig 3 shows a schematic of the stimuli used – each sequence comprised 30 tones in repeated LHL- triplets, where the dash represents a silent gap. Each tone was 50 ms in duration, including 10 ms raised cosine onset and offset ramps. A $4 \times 5$ factorial design was used: the pitch of the high tones took values of 3, 6, 9, 12, and 15 semitones above the low tone, which had a fixed frequency of 1000 Hz, and the offset to onset interval took values 17, 33, 50, and 67 ms. The duration of the silent gap was equal to the tone duration plus the offset-onset interval. Conditions were ordered randomly – each condition was tested 20 times over 4-6 runs, each run lasting approximately 7 minutes. At the end of the sequence participants pressed a key to report whether the percept at the end of the sequence was most like a single stream (a galloping rhythm) or two separate streams of notes. The whole experiment lasted about 50 minutes.

**Experiment 2.** Twenty-six participants were enrolled for this study, all under-graduate students at Durham University Psychology programme. No personal data was kept, ensuring participants' anonymity.

**Material and stimuli.** Each testing trial consisted in 2 sequences of 4 pure tones in repeated Low-Medium-High-Medium (L-M1-H-M2 or L-M2-H-M1) quadruplets. The first sequence was always repeated 22 times for a total of 88 tones presented. The second one was always repeated a total of 11 times for a total of 44 tones presented. Tones between sequences within a same trial were always the same and only their order of presentation could differ. Each tone was 100ms in duration, including 10ms raised cosine onset and offset ramps. The offset to onset interval between tones inside a sequence was 16.67ms. Each sequence also had general 500ms long raised cosine onset and offset ramps. The offset to onset interval between sequences inside a trial was 2s. The lowest tone had a fixed frequency of 440Hz across trials. The highest possible tone had a frequency of 2960Hz. The lowest frequency was specifically chosen to correspond to a common tone, and to control for differences in perceived loudness as much as possible across the range of played frequencies. Indeed, the 440-2960Hz range presents a low variability in equal-loudness [59] The frequency of M1 was calculated in semitone increases from the lowest tone, according to experimental conditions. M2 was always 3 semitones higher than M1. As was the case for the difference between L and M1, the frequency of H was calculated in semitone increases from M2, in relation to experimental conditions (see Fig 9 for a representation of a typical trial).

Training trials were similarly designed and consisted in 2 sequences of 3 pure tones in repeated Low-Medium-Medium (L-M1-M2 or L-M2-M1) triplets.

**Design.** A $2 \times 6$ within-subjects factorial design was used. The first independent variable was the medium tones inversion, which could be either present or absent between the 2 sequences of a trial. This resulted respectively in trials objectively comprised of a pair of different sequences, and trials objectively comprised of a pair of twin sequences. The second independent variable was pairs of frequency differences between L and M1, and between M2 and H, counted in semitones. Possible values were 3-3, 3-9, 3-15, 9-9, 9-15, and 15-15. Conditions were presented randomly, and the order in which baseline and inverted tone sequences were presented was counterbalanced. Each identical pair of sequences was presented 6 times, while each different pair of sequences was presented 12 times for a total of 108 trials. Training trials consisted in 3 identical pairs and 3 different pairs repeated twice, for a total of 12 training trials.

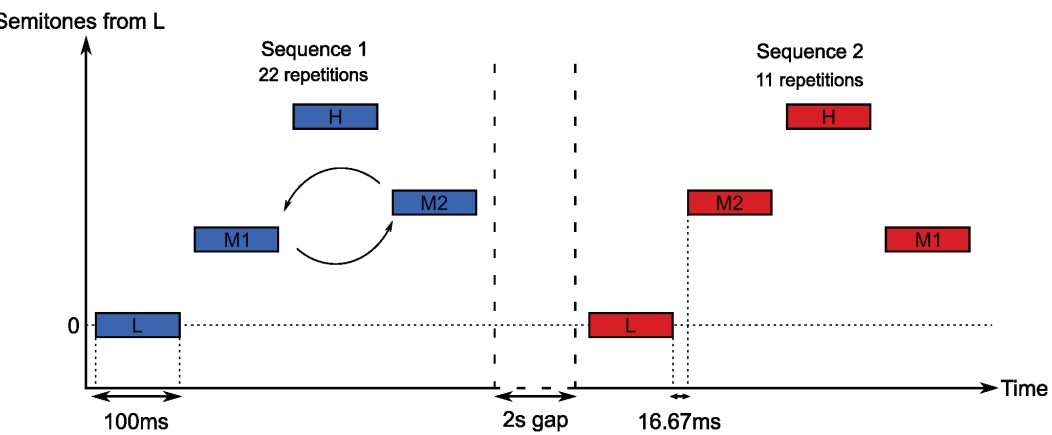

**Fig 9. Visual representation of a trial with inversion in a 9-9 frequency difference condition.**

The first dependent variable was the perceived difference between sequences (different vs. similar). The second dependent variable was the level of confidence in this judgement (on a scale from 1 to 4, 4 being "very confident"). Confidence judgments were included as a purely exploratory variable, and preliminary analyses did not reveal any meaningful insights. They were therefore dropped from the manuscript.

**Procedure.** Participants were greeted in a small sound-attenuated room and were asked to sit at a desk, approximately 60cm from a computer screen. They were handed an information and a privacy notice sheet, stating the general aims of the experiment, their rights as a subject, and how their data would be handled. After reading and asking any questions they may have to the experimenter, they were asked to sign a consent form. They were then asked to put the headphones on once they confirmed they understood the task instructions.

Participants were asked to listen to pairs of melodies presented sequentially, and judge after each pair if the melodies were similar or different by pressing the right key on a keypad ("1" for different, "2" for similar). They were also asked to rate their confidence about the judgement they just made, by pressing a key from 1 to 4 on the same keypad. Participants were warned that tones between sequences had the same frequency, and that they should focus on the order of tones within the melody. On each trial, a white dot was displayed in the middle of the screen for a brief period to signify a new trial was about to start. A grey dot was then displayed in place of the white one along with the instruction "listen" while melodies were being played. A black dot, along with a reminder of the response keys, replaced them as soon as the melodies were finished, meaning that subjects could enter their response. Participants had no time limit to respond, as the next trial would only start after they did.

Once the experiment was over, the experimenter gave an oral feedback explaining the aims, design, and experimental background of the study. The whole experiment lasted about 55 minutes.

## Supporting information

**S1 Text. Supporting figures, parameter recovery, and model recovery.**
(PDF)

## Author contributions

**Conceptualization:** Nathanael Larigaldie, Tim Yates, Ulrik R. Beierholm.

**Formal analysis:** Nathanael Larigaldie, Ulrik R. Beierholm.

**Investigation:** Nathanael Larigaldie, Tim Yates.

**Methodology:** Nathanael Larigaldie, Tim Yates, Ulrik R. Beierholm.

**Software:** Nathanael Larigaldie, Tim Yates, Ulrik R. Beierholm.

**Supervision:** Ulrik R. Beierholm.

**Writing – original draft:** Nathanael Larigaldie, Tim Yates, Ulrik R. Beierholm.

**Writing – review & editing:** Nathanael Larigaldie, Ulrik R. Beierholm.

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
