## [Decision Letter · Decision Letter 0]

4 Dec 2024

PCOMPBIOL-D-24-01574

Perceptual clustering in auditory streaming

PLOS Computational Biology

Dear Dr. Beierholm,

First of all, my apologies for the delay in processing your submission. It took me a while to secure a third reviewer and then the reviews came back in a week where I was extremely busy myself.

Thank you for submitting your manuscript to PLOS Computational Biology. Based on the reviews of myself and 3 independent reviews, we feel that it has merit but does not fully meet PLOS Computational Biology's publication criteria as it currently stands. Therefore, we invite you to submit a revised version of the manuscript that addresses the points raised during the review process.

As you will see below, the reviewers are largely in agreement about both the merits and shortcomings of the manuscript. In summary, they indicate that the manuscript mainly requires clarifications, in several areas:

1. Mechanistic implementation - two of the reviewers indicate that the model is likely too complex to be implemented by human brains, which warrants some discussion about the biological counterpart of your computational-level model

2. Mathematical presentation - the notation can be improved at a few places and some equations were hard to read

3. Modelling assumptions - two of the reviewers wonder about how realistic some of the assumptions are, which warrants some discussion

4. Presentation of figures, tables, and results - the reviewers point out several points at which the presentation can be improved; one of them suggests to focus on intuition-building, which I believe is a great suggestion

For a more details overview of the comments, see the reviews attached below.

Please submit your revised manuscript within 60 days Feb 03 2025 11:59PM. If you will need more time than this to complete your revisions, please reply to this message or contact the journal office at ploscompbiol@plos.org. Please include the following items when submitting your revised manuscript:

We look forward to receiving your revised manuscript.

Kind regards,

Ronald van den Berg

Academic Editor

PLOS Computational Biology

Hugues Berry

Section Editor

PLOS Computational Biology

Feilim Mac Gabhann

Editor-in-Chief

PLOS Computational Biology

Jason Papin

Editor-in-Chief

PLOS Computational Biology

**Journal Requirements:**

At this stage, the following Authors/Authors require contributions: Nathanael Larigaldie, Timothy A Yates, and Ulrik R. Beierholm. Please ensure that the full contributions of each author are acknowledged in the "Add/Edit/Remove Authors" section of our submission form.

3) Thank you for including an Ethics Statement for your study. Please include:

i) A statement that formal consent was obtained (must state whether verbal/written) OR the reason consent was not obtained (e.g. anonymity). NOTE: If child participants, the statement must declare that formal consent was obtained from the parent/guardian.].

5) Please ensure that all Figure files have corresponding citations and legends within the manuscript. Currently, Figure 8 in your submission file inventory does not have an in-text citation. If the figure is no longer to be included as part of the submission, please remove it from the file inventory.

6) We notice that your supplementary information is included in the manuscript file. Please remove them and upload them with the file type 'Supporting Information'. Please ensure that each Supporting Information file has a legend listed in the manuscript after the references list.

**Reviewers' comments:**

Reviewer's Responses to Questions

Reviewer #1: Review of "Perceptual clustering in auditory streaming"

Reviewer: Michael Landy

This is a nice paper on a model of causal inference that sensibly extends to stimuli with more than two items. The original CI model from Koerding and colleagues was based on only a pair of stimuli and a generative model with only two scenaries (same-source and separate-sources). As the authors know, I and many others have published reams of papers based on the dichotomous setup. In fact, I have toyed with a model for scenarios with more than two items (i.e., multiple flashes and beeps), and the combinatorics made for very clunky modeling (and I've never finished nor published the project). So, this straightforward bit of modeling is a welcome addition. I have a bunch of comments, but all in all I very much like this paper.

Specifics:

* Eq. 1: Some variations and clarifications here are worth a mention. Delta-t here is ISI, and I imagine that ISI and SOA might both matter, but that gets into the complexities of rhythm perception, and the experiments here all avoid any need for that, so it's fine as is. Delta-f is never clearly defined. One might imagine f is in Hz, but all Methods describes them in log scale (multiples of a semitone), so you might say that this is a Gaussian in something like "log-frequency velocity". I only know superficial things about the stream-segregation literature. Is log-frequency difference the right variable here (independent of base frequency)? Does stream segregation also take into account the circular nature of pitch perception, so that octave leaps as LESS likely to segregate than small leaps?

* 122-145: I had some trouble plowing through this section to get a clear handle on what was being assumed and whether it justified all the simplifications here. A clear generative model would have been helpful (possibly with a model diagram). When you apply Bayes Rule to obtain the 2nd line of Eq. 5, you make some kind of conditional-independence assumption. You also say that just above Eq. 5, but it's unclear exactly what is conditionally independent of what, nor how strong (and acceptable) an assumption that might be. True, the denominator in that 2nd line can be figured out as whatever normalizing constant needed. In any case, some clarification along the way would be helpful. Note that in the first and 2nd lines of Eq. 5 you have "S_2,S_3" just after a condition "|", when in fact you mean "S_1,S_2".

* Figure 6: You should label the color bar in (a) (yes, it's in the legend, but it's nice to be able to look at a figure and understand it as is).

* 288: I'd avoid obscure acronyms like "ASA".

* 300: "If medium tones were to be part...". I totally tripped over this sentence and couldn't understand why UNTIL I noticed the obscure text at the top of Figure 9 that talked about the 22 and 11 repetitions. This repetition is never described in the running text and even less clear, but important, is that during the repetitions, the SOA from M1 to M2 and the SOA from M2 to the next repetition's M1 are identical. That's why an inversion is hard to detect. Of course, potentially the participant could discriminate the two sequences by noting which tone came last, but they didn't know it was an integer number of repeats. So, you should clarify all of this somewhere.

* Table 1: The labels of the rows as "Different" and "Similar" is unclear. What you mean is that these are rows for stimuli in which the two sequences were identical vs. inverted. Maybe you should label the rows as Hit Rate and False-Alarm Rate and clarify in the legend.

* Figure 8 and text above: Yes, it's pretty disconcerting that ELBO was at chance or very close to it for about half your subjects (and all models including yours). The figure should repeat the key for the colors from Figure 6.

* Discussion: This is still a very fancy model to computer and one wonders whether the brain perhaps does something more heuristic and simpler. Perhaps some kind of greedy algorithm that collects evidence and keeps around only a single interpretation or a very small number, plus confidence, accumulating evidence without keeping lots of hypotheses around.

* Eq. 8: You are using 1 and 0 to represent responses of 1 and 2, which seems unnecessarily confusing.

* 494: The alternative to a single-stream hypothesis does not have to be half-half.

* Eq. 11 and surrounding text: This supports the 2-stream response even if there's a high probability of 3 or more streams.

Reviewer #2: The authors develop and test a novel Bayesian model that infers the causal structure of sound sources in auditory streams. Specifically, they reformulate auditory stream segregation as a Bayesian clustering problem: They develop a Bayesian model with a non-parametric prior based on a Chinese restaurant process (CRP) that allows updating of posterior distributions over potential causal structures with potentially infinite number of sources, only inferred from previous sensory inputs.

Overall, by assuming a CRP, the model elegantly infers causal structure of complex sound sources, going beyond previous causal inference models that only take two potential causal structures into account. The model qualitatively replicates important effects in auditory stream segregation (e.g. effects of tone frequency and ISI and the Gallopping effect), and is quantitatively fitted to a replication experiment of the Gallopping effect and a novel experiment testing a novel model prediction (i.e., the effect of melody segmentation on melody discrimination). The manuscript is well written and comprises an important novel modelling approach for a fundamental problem in auditory perception (causal inference on complex sound source structures in auditory streams). Importantly, the CRP model could be generalized to perceptual problems of causal inferences in other sensory modalities and specifically multisensory perception, so the work is very innovative and has wide applicability as well as relevance. Yet, I have some comments and concerns on the modelling approach, the experimental data and the presentation of results:

Intro

• L. 107 “the probability”, this definition requires further explication: Which probability is meant here exactly, I assume a likelihood for tone frequency and temporal distance, conditioned on the same source as the previous tone? The argument that the frequency of a tone from the same source cannot change instantly is plausible (and therefore frequency distance decreases the probability), but why does “the probability” increase with a longer ISI between previous tone offset and onset of the current tone? This counterintuitive assumption is presumably need to model the effect of presentation speed in Fig. 3a/b, but the rationale is unclear. From natural sound statistics, I would assume the reverse: longer pauses between two tones make it less likely that they have the same source (i.e., a new source is more likely).

• L. 110: “We here assume that the observer has a perfect noise free access to the generated auditory frequencies.” This is not a realistic assumption in any sensory system, how would the model be complicated by adding sensory noise? Or more generally, a hallmark of Bayesian models of perception is that they take sensory uncertainty into account, which effects would the model predict for e.g. low vs. high sensory noise in frequency and time estimation?

• Equation (2): The alpha parameter is undefined, I assumed it is meant as a small probability? In general, the Chinese restaurant process / Dirichlet process could be explained to increase readability in the Introduction.

• Fig. 2 and equation (6): The likelihood for a tone from a new source is highest when it is closed to f0, i.e. the average or midpoint of frequency of previous tones in a trial. This definition of f0 is counterintuitive to me: I would rather expect that a new source is likely if the frequency is very different from previous frequencies (i.e., higher or lower). Clustering of frequencies around a mean/midpoint is rather likely if the same source emits more tones.

Results, Exp. 1 / 2:

• Fig. 6 gives a somewhat incomplete impression of the behavioral and modelling results: Why is the model prediction only shown for subject 8? Why is not the average data shown for 15 participants but only 7 selected participants? Showing mean data and mean model predictions for all participants (or behahavioral data / model predictions for several subjects side-by-side) would give a more comprehensive impression. What is the overall model fit of the CRP model (e.g. R^2) for Exp. 1 and 2?

• The main CRP model is compared to competing models with simpler ad-hoc priors in Exp. 1/2. However, how does the CRP model compare to models which use different non-optimal/heuristic decision strategies, such as stochastic sampling of same source / new source for each tone or deciding for same / new source based on a fixed threshold on pitch increment and ISI (cf. Acerbi et al., 2018 Plos Comp Biol)?

• Exp. 2: The definitions of frequencies such as 3-3 (L. 326) is unclear at this point of the manuscript, frequency difference between low-middle, or middle-high tones? It is defined in the methods, but a brief definition here would help the reader. Even though the predictions for Exp. 2 are nicely explained, it remains underexplained how this translates into d prime differences. I.e., the readability for exp. 2 could be enhanced.

Discussion

• The authors discuss the effect that later tones can retroactively change the causal attribution of previous tones. This is actually a very interesting consequence of causal inference from accumulating evidence, and could be more broadly discussed in the context of postdiction in perception (e.g., Choi & Scholl 2006; Stiles, Tanguay, Shimojo, 2021).

• The model accounts for ‘causal uncertainty’ about the generating structure. As mentioned above, further uncertainty from sensory noise is a more realistic assumption. How could this be incorporated into the model, and how would it change model behaviour (e.g. prediction of novel effects?)? The authors claim that the model is very general, e.g. “the framework allows more complex cues from audition and other modalities to be used as long as their perceptual difference can be quantified” (L. 89): I imagine that e.g. adding visual cues with uncertain temporal structure and integration of audiovisual cues based on relative precision could critically depend on modelling sensory noise. In other words, while mere clustering of stimuli into a Gestalt may neglect sensory noise, it becomes crucial when stimuli are not only grouped, but integrated or unified into a single percept.

• For an example of causal inference / clustering in the visual modality, the authors may also discuss causal inference for visual motion perception (e.g., Shivkumar, DeAngelis, Haefner 2023)

Methods

• The methods do not explain how the model predictions for replicating previous auditory stream segregation effects (Fig. 3/4) were generated.

• For the sake of completeness/reporting standards, the authors may report some basic demographics (age, sex distribution) of the sample (if available).

• Exp. 2:

o Why was the different sequences presented in twice as many trials as the identical sequences, and not with equal proportion? This could have introduced a response bias towards a “different” response.

o Participants also provided confidence ratings in each trial. Did this data reveal any meaningful pattern regarding auditory stream segregation / the modelling approach?

• Parameter recovery could be accompanied by model recovery for the CRP and competing models (relevant esp. for a novel model; cf. Wilson & Collins 2019 eLife).

The authors make all data and code available on OSF.

Reviewer #3: This paper brings an existing mathematical tool (nonparametric Bayesian clustering) to bear on the problem of modeling causal inference in the brain when there exist a potentially unlimited number of sources, focusing specifically on the case of auditory streaming. The key insight is that an existing peg fits an existing hole, and indeed it does. Most importantly, the authors do substantial legwork to demonstrate the quality of the fit by showing that this mathematical formalism naturally accounts for a range of existing (and one new) auditory psychophysics results, and that it can easily be fit to individual data, outperforming several alternative models. The goal of the paper is successfully accomplished, and the insight is interesting enough to merit publication.

One minor qualm I have with the argument here is that even though the generative model can now account for unlimited sources, we have only kicked the can down the road -- we still need a posterior approximation that makes the computations tractable. These approximations are not just mathematical conveniences -- they correspond to real strategies that the brain might employ to utilize limited resources, and they make behavioral predictions. This point need not be fully explored in this paper, but it deserves mention in the main body of the text and not relegation to the appendix.

This is part of the larger question of mechanistic implementation of the algorithm by the wetware, which the authors should acknowledge is left for future work. When the authors do mention mechanistic models, it is only for the purpose of evaluating them in competition with the current model. They should make it clear that their model is in want of a mechanistic implementation, and when appropriate should evaluate whether features of existing mechanistic models might help to fill this void. Mentioning Marr's Levels is not mandatory but could be useful.

I am a firm believer in devoting text in modeling papers to intuition-building. When the authors list perceptual phenomena accounted for by their model, I would like to see them offer some intuitive explanation for why their approach accounts for these observations. For me this was particularly an issue in the discussions of stream segregation following the accumulation of sound pattern repetitions (starting on line 197) and the bouncing percept (213), key strengths of the model. Please explain why the model does these things!

Other minor complaints:

- The Galloping Effect section is strange after the section on Time and Frequency which already explained and invoked the galloping effect. Maybe there's a better way to organize this material?

- L12: "Inferring the generative model" is awkward language. Usually one would say "inferring the parameters of the generative model," though I see why this might not make sense for a "nonparametric" model.

- The "3-3" notation is used extensively in the presentation of Experiment 2 but is not well explained in the body text.

- L34: should be "renowned"

- Inappropriate commas on lines 16, 29, 31

- "i.e." should be "e.g." on line 16

- L117: awkward notation -- why is n not indexed with the candidate source j? I think maybe this should say:

If the number of tones previously assigned to source s prior to event i is given by n_s^i = /sum_{k=1}^{i-1} delta( s - S_k ) , then

p( S_i = s | ...

[etc... or you could leave out the equation for n_s^i since it's obvious]

- Missing comma on 353

- L491 and below: is r_k drawn from {0,1} or {1,2}?

- L498 -- I would find it helpful here to specify which parameters were fit for each model (one of which is beta, which you just introduced) -- I think this only becomes fully clear in the Supporting Information.

**Have the authors made all data and (if applicable) computational code underlying the findings in their manuscript fully available?**

Reviewer #1: **No: **

Reviewer #2: Yes

Reviewer #3: Yes

PLOS authors have the option to publish the peer review history of their article (what does this mean?). If published, this will include your full peer review and any attached files.

Reviewer #1: No

Reviewer #2: **Yes: **Tim Rohe

Reviewer #3: **Yes: **Jonathan Cannon

**Figure resubmission:**
---

## [Decision Letter · Decision Letter 1]

13 Mar 2025

PCOMPBIOL-D-24-01574R1

Perceptual clustering in auditory streaming

PLOS Computational Biology

Dear Dr. Beierholm,

Thank you for submitting your manuscript to PLOS Computational Biology. 

Your revised manuscript was reviewed by myself and the same three reviewers as in the first round. As you can see in the comments below, the reviewers were generally happy and had only very minor comments left. Except for one: reviewer #2 was disappointed to see that their suggestion to include a model recovery analysis was not followed up on. The reviewer argues that it is good practice to do such an analysis, regardless of whether the presented models are or variants of each other or not. I agree with that view - model comparison results on empirical data are easier to interpret when accompanied by a model recovery analysis using the same methods (and similar number of trials and participants in the synthetic data). Since this is likely a rather straightforward for exercise for you, I would very much like to see this included in the final revision.

Since only minor revisions are required, I will probably be able to make a final decision on your next submission without having to send it back to the reviewers.

Please submit your revised manuscript within 30 days May 13 2025 11:59PM. If you will need more time than this to complete your revisions, please reply to this message or contact the journal office at ploscompbiol@plos.org. Please include the following items when submitting your revised manuscript:

We look forward to receiving your revised manuscript.

Kind regards,

Ronald van den Berg

Academic Editor

PLOS Computational Biology

Hugues Berry

Section Editor

PLOS Computational Biology

**Reviewers' comments:**

Reviewer's Responses to Questions

Reviewer #1: Re-review of "Perceptual clustering in auditory streaming"

Reviewer: Michael Landy

I liked this paper the last time, and I still do. Not a lot of comments:

Line 164: Isn't this simply another way of specifying \alpha (from Eq. 2)? Why isn't \alpha mentioned here?

170: This text should refer back to Eq. 1

You use \sigma in Eq. 1, where it is in units of semitones/sec, and again in Eq. 6, where it's in units of semitones. These can't be the same \sigma, and yet you act like you have only a single \sigma parameter. Clarify. In line 310 it says it's the former (semitones/sec).

Fig. 2: Don't you need to specify that \alpha is forced to be less than one (to keep the new cluster with the smallest prior probability)?

196: a verb is missing here

Fig. 3: These dendrograms don't really cluster the tones exactly as one would wish. In (b) one would expect 2/4/6 to be grouped. In (c) it should be 123 and 456. Thoughts?

289: The supplement does NOT have the individual subject data and fits.

Reviewer #2: The authors address most of my concerns very well, expect for the model recovery. Overall, the manuscript has improved, especially in readability, and the paper is an important contribution.

My point-by-point responses:

• Definition of probability of a common source: I agree with the authors that the probability for a common source depends on temporal distance between two tones on different time scales. I still the explanation in the manuscript too short, while the explanation in the response latter is clearer: If a sound source continues oscillating at the same frequency, two consecutive tones of the different frequency are unlikely while two tones (at the same frequency) with larger temporal distance are likely. I still find it difficult to follow their argument, maybe they could explain a bit more elaborately in the manuscript. An option would also be to plot the normal distribution (equ. 1) to give the reader a more intuitive understanding (e.g. supplemental).

• Effect of adding sensory noise:

• Alpha parameter and Chinese restaurant process: These concepts have now been clarified, even though the CRP could be intuitively (very briefly) explained in the Intro to increase readability.

• I agree that for a prior distribution for a new source assumes that the tone will still be in the range of previous frequencies (as a prior guess).

• Fig. 6/7: The authors now show model comparison results of the whole sample (with clear support for the CRP model in each participant in Exp.1 and most participants in Exp.2), a measure of overall model fit, and model predictions for more participants in the main text. The authors promise to show model fits from all participants for Exp. 1 in the supplemental, but unfortunately I do not find it there (only the graphical model and parameter recovery)??

• Alternative decision strategies: The authors now discuss the possibilities of alternative decision strategies, leaving it to future research to investigate this, which is fine. One point: “Any subjects using a fixed threshold in either pitch increment or ISI would be easy to detect in the data in Exp 1 as they would show up as vertical or horizontal lines/boundaries in FIg. 6b.” 6B shows the ELBO values, I guess the authors refer to 6A (and in the promised supplemental figures), where indeed a fixed-threshold strategy would show as a clear horizontal/vertical pattern.

• The authors now sufficiently explain the definitions of frequencies in Exp. 2 and the computation of d prime.

• In the Discussion, the authors added the relation to postdiction, the effect of modelling perceptual noise and applying the model to multisensory causal inference (some refs are missing here), and visual motion clustering.

• Basic demographical data cannot be reported because this data was not recorded. I think it is good scientific practice to report at least minimal data on human participants, but for a theoretical contribution this might be less important.

• For Exp. 2 , the authors comprehensively justify the imbalance of different/same sequences and why they omitted the confidence ratings.

• The authors do not include a model recovery analysis, arguing that their competing models are only variants of the CRP model. I disagree to that point: Even if the authors use similar models as valid possible alternatives, it still makes sense to check whether model fitting / model analyses pipelines can reliably recover the different models. This is a question of methodological rigor, not a question of whether the competing models are old, new or a special case of the main model (which often holds for simpler competing models). Thus, I still think that model recovery is essential for a modelling paper introducing a novel computational model.

Reviewer #3: The authors have responded satisfactorily to my comments. However, they have introduced some text that is confusing.

Authors write:

"By default, the model is biased towards the most parsimonious interpretation of events (Occam’s razor: as little different causes as possible)"

Parenthetical is awkward and grammar is off. How about:

"By default the model is biased towards the most parsimonious interpretation of events (Occam's razor), which in this case means creating as few different causes as possible."

A paragraph starts with the topic sentence "Importantly, the model goes beyond giving just the number of sources, but says which sounds are produced by each source." They have now added sentences to the end of the paragraph about alternative strategies -- I don't see how this fits the paragraph or relates to the previous sentence.

"a very low number of the most plausible (or even a single) interpretations in memory." A model that maintains only one interpretation can't postdict. I think authors should cut "(or even a single)".

Paragraph at Line 460: Authors say that THEIR use of simplifications/approximations suggests that the BRAIN, which draws on a larger stream of perceptual data, may use even simpler approximations. This reasoning doesn't seem quite right to me. Isn't it simply the case that ANY system trying to implement this algorithm must either deal with an exploding number of possible interpretations or make substantial simplifications to the algorithm? The authors had to contend with that in their program, and the brain has to contend with that. Unless the authors do not agree with this analysis, I think that is what they should say.

**Have the authors made all data and (if applicable) computational code underlying the findings in their manuscript fully available?**

Reviewer #1: Yes

Reviewer #2: Yes

Reviewer #3: Yes

PLOS authors have the option to publish the peer review history of their article (what does this mean?). If published, this will include your full peer review and any attached files.

Reviewer #1: **Yes: **Michael S Landy

Reviewer #2: No

Reviewer #3: **Yes: **Jonathan Cannon

**Figure resubmission:**
---

## [Editor Report · Decision Letter 2]

3 Jun 2025

Dear Dr. Beierholm,

We are pleased to inform you that your manuscript 'Perceptual clustering in auditory streaming' has been provisionally accepted for publication in PLOS Computational Biology.

Best regards,

Ronald van den Berg

Academic Editor

PLOS Computational Biology

Hugues Berry

Section Editor

PLOS Computational Biology

---

## [Editor Report · Acceptance letter]

PCOMPBIOL-D-24-01574R2

Perceptual clustering in auditory streaming

Dear Dr Beierholm,

I am pleased to inform you that your manuscript has been formally accepted for publication in PLOS Computational Biology. Your manuscript is now with our production department and you will be notified of the publication date in due course.

With kind regards,

Judit Kozma
